# Multimodal Treatment in Metastatic Colorectal Cancer (mCRC) Improves Outcomes—The University College London Hospital (UCLH) Experience

**DOI:** 10.3390/cancers12123545

**Published:** 2020-11-27

**Authors:** Nalinie Joharatnam-Hogan, William Wilson, Kai Keen Shiu, Giuseppe Kito Fusai, Brian Davidson, Daniel Hochhauser, John Bridgewater, Khurum Khan

**Affiliations:** 1Department of Gastrointestinal Oncology, University College London Hospital NHS Foundation Trust, London NW1 2PG, UK; n.joharatnam@nhs.net (N.J.-H.); kaikeen.shiu@nhs.net (K.K.S.); d.Hochhauser@ucl.ac.uk (D.H.); j.bridgewater@ucl.ac.uk (J.B.); 2Department of Statistics, Cancer Research UK and UCL Cancer Trials Centre, London W1T 4TJ, UK; william.wilson.13@ucl.ac.uk; 3HPB and Liver Transplant Unit, Royal Free London NHS Foundation Trust, London NW13 1QG, UK; g.fusai@nhs.net (G.K.F.); b.davidson@ucl.ac.uk (B.D.); 4Research Department of Oncology, UCL Cancer Institute, London WC1E 6BT, UK; 5Department of Medical Oncology, North Middlesex University Hospital, London N18 1QX, UK; 6Department of Oncology, GI Cancer Lead North London & Oncology Research Lead North Middlesex University Hospital, London N18 1QX, UK

**Keywords:** colorectal cancer, colorectal liver metastases, radiofrequency ablation (RFA), Microwave ablation (MWA), stereotactic body radiation therapy (SBRT), selective internal radiation therapy (SIRT)

## Abstract

**Simple Summary:**

Most patients with metastatic colorectal cancer (mCRC) are treated with chemotherapy alone; however, increasingly, addition procedures such as surgery, radiotherapy, and ablation of metastases are being used with the aim of improving survival. We aimed to review all patients with mCRC receiving chemotherapy at the University College London Hospital (UCLH) in order to determine the survival outcomes of those who received additional procedures versus those who received chemotherapy alone, but also to understand which procedures provided the most benefit and to whom. We found that additional procedures improved survival of patients with mCRC compared to chemotherapy alone, and the more procedures performed, the greater the survival benefit. Surgical removal of metastases were associated with the most improved survival; however, repeated procedures in the same organ did not improve outcomes. We hope these results may help guide cancer specialists in the decision-making process for the management of patients with mCRC.

**Abstract:**

Background: Despite notable advances in the management of metastatic colorectal cancer (mCRC) over the last two decades, treatment intent in the vast majority of patients remains palliative due to technically unresectable disease, extensive disease, or co-morbidities precluding major surgery. Up to 30% of individuals with mCRC are considered potentially suitable for primary or metastasis-directed multimodal therapy, including surgical resection, ablative techniques, or stereotactic radiotherapy (RT), with the aim of improving survival outcomes. We reviewed the potential benefits of multimodal therapy on the survival of patients with mCRC treated at the UCLH. Methods: Clinical data on baseline characteristics, multimodal treatments, and survival outcomes were retrospectively collected from all patients with mCRC receiving systemic chemotherapy between January 2013 and April 2017. Primary outcome was the impact of multimodal therapy on overall survival, compared to systemic therapy alone, and the effect of different types of multimodal therapy on survival outcome, and was assessed using the Kaplan–Meier approach. All analyses were adjusted for age, gender, and side of primary tumour. Results: One-hundred and twenty-five patients with mCRC were treated during the study period (median age: 62 years (range 19–89). The liver was the most frequent metastatic site (78%; 97/125). A total of 52% (65/125) had ≥2 lines of systemic chemotherapy. Of the 125 patients having systemic chemotherapy, 74 (59%) underwent multimodal treatment to the primary tumour or metastasis. Median overall survival (OS) was 25.7 months [95% Confidence Interval (CI) 21.5–29.0], and 3-year survival, 26%. Univariate analysis demonstrated that patients who had additional procedures (surgery/ablation/RT) were significantly less likely to die (Hazard Ratio (HR) 0.18, 95% CI 0.12–0.29, *p* < 0.0001) compared to those receiving systemic chemotherapy alone. Increasing number of multimodal procedures was associated with an incremental increase in survival—with median OS 28.4 m, 35.7 m, and 64.8 m, respectively, for 1, 2, or ≥3 procedures (log-rank *p* < 0.0001). After exclusion of those who received systemic chemotherapy only (*n* = 51), metastatic resections were associated with improved survival (adjusted HR 0.36, 95% CI 0.20–0.63, *p* < 0.0001), confirmed in multivariate analysis. Multiple single-organ procedures did not improve survival. Conclusion: Multimodal therapy for metastatic bowel cancer is associated with significant survival benefit. Resection/radical RT of the primary and resection of metastatic disease should be considered to improve survival outcomes following multidisciplinary team (MDT) discussion and individual assessment of fitness.

## 1. Introduction

Both globally and in the United Kingdom, colorectal cancer (CRC) is the second leading cause of cancer mortality [1,2]. Around 25% of patients have metastatic (stage IV) disease at the time of diagnosis, with five-year overall survival for these patients as low as 10% [3], and median overall survival in patients with unresectable disease approximately 30 months with systemic treatment [4,5].

The majority of patients with metastatic CRC (mCRC) are treated using systemic chemotherapy with palliative intent. However, increasingly a multimodal treatment approach is being adopted, using surgery, ablation, or radiotherapy to improve disease control, with the hope of survival benefit and possibly cure [6,7].

Five-year survival rates following resection of hepatic metastases have been reported to be as much as 40–60% [8]. Local treatment approaches, including surgery or ablation, are usually recommended in patients with oligometastatic disease involving the primary and limited spread to one of the common sites of metastases, such as the liver, lung, peritoneum, ovary, or nodes [5]. Patients with metastases to the bone or brain have a poor prognosis, and local treatments are considered less effective [9,10]. Similarly, in patients with extensive multiple organ metastases, a multimodal approach is more controversial [11]. In patients in whom surgical resection is not considered feasible, other local ablative techniques may still be of value, including radiofrequency ablation (RFA), microwave ablation (MWA), stereotactic body radiation therapy (SBRT), and selective internal radiation therapy (SIRT), which may improve disease-free interval, allowing a break from systemic treatment [5].

A review of more than 82,000 patients with metastatic colorectal cancer using data from the U.S. National Cancer Database found that the cohort undergoing bowel resection and lung resection had the best overall survival (5 year overall survival (OS) of 44.5%) of those with metastatic disease, compared to primary resection and liver resection and primary with both lung/liver resections (5 year OS 35.2% and 20.1%, respectively; *p* < 0.001) [12].

Numerous case studies have shown that long-term survival can be achieved in 20–50% of patients who undergo resection or ablation of metastases [5,11]. Local and metastases-directed treatments should be considered following multidisciplinary team (MDT) discussions, with therapy based on site of disease, the goal of treatment (with an aim for R0 resection or complete ablation), and patient fitness [13]. Most studies have evaluated outcomes from single techniques in single site metastatic disease. Studies deriving data from population databases often have limited details of individual patients, and hence analysis of variables affecting outcomes for multi-disciplinary treatment may not be possible. Real world data on patients with mCRC in the United Kingdom is limited, but helps to provide insights for clinicians into treatment patterns and outcomes in clinical practice. Therefore in this paper, we report a retrospective analysis of the use of and survival outcomes from multimodal therapy in patients with metastatic colorectal cancer at a large teaching hospital, University College London Hospital (UCLH), over a 4-year period, providing a real-world review of practice and outcomes.

## 2. Results

### 2.1. Patient Characteristics

From January 2013 to April 2017, a total of 151 patients were treated for metastatic colorectal cancer in the medical oncology clinic; 26 patients were excluded from further analysis secondary to incomplete follow-up data, transfer to another centre, or being treated with best supportive care alone. Median follow-up in the study was 54 months. The remaining 125 patients with mCRC treated during the study period had a median age of 62 years (range 19–89), of which 18% (22/125) were ≤50 years; 56% (70/125) were male.

The majority of primary tumours were left-sided (66%; 82/125), with rectum/rectosigmoid the most prevalent primary site (38%). There were nine cases (7%) of mucinous adenocarcinoma and two cases of signet ring adenocarcinoma (1.6%). Of the study population with available mutational analysis results (82%; 102/125), 53% (54/102) were Ki-ras 2 Kirsten rat sarcoma (KRAS) wild type (WT) and 10% B-Raf proto-oncogene, serine/threonine kinase (BRAF) mutant (10/96), and 7% of those with availability of mismatch repair (MMR) status (5/71) were deficient, consistent with the prevalence in the literature [14]. Those with MMR deficiency were predominantly right-sided primary tumours (80%), and three had peritoneal disease (60%).

The liver was the most frequent site of metastatic disease, affecting 78% (97/125), and liver-only metastases were present in 44% (55/125). Lung metastases were present in 26% (33/125), and the lung was the only site of metastatic disease in 7% (9/125). A total of 47% (59/125) had the primary in situ at the end of the study period.

### 2.2. Systemic Therapy

Of the study population, 34% received either neoadjuvant and/or adjuvant chemotherapy (42/125), and 18% (23/125) received adjuvant chemotherapy only. A total of 30 of these patients received neo/adjuvant chemotherapy for primary surgery first, and 8 for metastases-directed surgery first (4 received chemotherapy with neoadjuvant intent, but subsequently progressed). Most patients (82%) received at least one line/course of palliative chemotherapy (102/125), with 37% (46/125) receiving targeted treatment as part of first-line therapy [either Vascular Endothelial Growth Factor (VEGF)- or Epidermal Growth Factor Receptor (EGFR)-directed], as per standard of care at the time. Of the population, 52% had ≥2 lines of systemic chemotherapy, and 19% received ≥4 lines.

A total of 41% (*n* = 51) of patients received systemic chemotherapy only.

### 2.3. Main Outcomes of Multimodal Therapy

Median OS for the entire patient group (*n* = 125) was 25.7 months (95% CI 21.5–29.0). One-year survival was 79%, and three-year survival was 26%. At the last follow up, 81% (101/125) had died, and the remaining 24/125 were alive and censored. Patient characteristics are summarised in Table 1.

Of the patients, 45% (56/125) were deemed to have curative resection options by the MDT at presentation, on the basis of technical resectability. A total of 59% (74/125) of patients underwent a primary tumour or metastasis-directed procedure. In this multimodal cohort of *n* = 74, 81% had primary directed treatment as the first multimodal intervention (primary surgery *n* = 55; primary RT *n* = 5).

Median OS for those who underwent any multimodal therapy (surgery/ablation/RT) (*n* = 74) was 33.6 months (95% CI 30.0–44.9), compared to those who had systemic chemotherapy alone (*n* = 51) at 14.4 months (95% CI 10.4–20.5). Univariate analysis showed that patients who had additional cancer procedures had better survival than those who received chemotherapy alone, adjusted for age, gender, and side of primary tumour (adjusted HR 0.18, 95% CI 0.12–0.29, *p* < 0.0001) (Figure 1).

Patients receiving increasing number of multimodal therapies also had improved survival. Median OS rates were 28.4 m, 35.7 m, and 64.8 m, respectively, for 1, 2, or ≥3 procedures (log-rank *p* < 0.0001) (Figure 2), regardless of type/site of procedure.

The survival of patients undergoing primary resection or primary radical RT was significantly improved when compared to the entire patient cohort (*p* < 0.001, adjusted for age, gender, side) (Table 2). However, after exclusion of those who received systemic chemotherapy only, on univariate analysis, primary resection/primary radical chemoradiotherapy was not significantly associated with improved survival (adjusted HR 0.50, 95% CI 0.23–1.08, *p* = 0.07). Similarly non-surgical metastasis-directed procedures (including drug-eluting bead trans-arterial chemoembolization (DEB-TACE), RFA, SBRT, and SIRT) showed a statistically improved survival when compared to the entire patient cohort (adjusted *p* < 0.003), however, they lost this survival benefit following exclusion for those receiving systemic therapy only (adjusted HR 0.66, 95% CI 0.33–1.32, *p* = 0.24). Surgical resection for metastases was associated with improved survival (adjusted HR 0.36, 95% CI 0.20–0.63, *p* < 0.0001), even after exclusion of patients who received systemic chemotherapy only.

### 2.4. Liver-Specific Outcomes

There was no evidence of a survival benefit in the whole cohort of mCRC patients in those with liver-only metastatic disease (HR 0.84, 95% CI 0.56–1.27, *p* = 0.40).

Excluding those individuals who received systemic chemotherapy alone (*n* = 51), 41 underwent a liver metastasis-directed procedure (surgical *n* = 33/ablative *n* = 8). There was no evidence of a survival benefit in those who had any type of liver procedure (HR 0.66, 95% CI 0.38 - 1.15, *p* = 0.145); however, in those who specifically underwent a hepatic resection, there was evidence of benefit (HR 0.54, 95% CI 0.30–0.97, *p* < 0.05) (Table 2). However, there appeared to be no significant advantage to undergoing multiple liver-directed procedures (*p* = 0.21) or multiple liver surgery (*p* = 0.13).

### 2.5. Lung-Specific Outcomes

Univariate analysis demonstrated that patients with lung-only metastatic disease had improved survival compared to those with other/multiple sites of metastases (adjusted for age, gender, side; HR 0.19, 95% CI 0.06–0.60, *p* = 0.005).

Excluding those patients who received only systemic chemotherapy (*n* = 51), undergoing any lung-directed procedure (HR 0.21, 95% CI 0.07–0.58, *p* = 0.003) or lung surgery (HR 0.16, 95% CI 0.04–0.66, *p* = 0.01) both demonstrated a significant survival advantage. However multiple lung-directed procedures had no benefit (HT 0.27, 95% CI 0.07–1.11, *p* = 0.07).

### 2.6. Multivariate Analysis

Multivariable analysis of the whole patient cohort (*n* = 125) including age, gender, side of primary, receipt of neo/adjuvant chemotherapy, presence of lung-only metastases, and receipt of primary surgery/RT and other non-surgical procedure consolidated the previous results that metastasis-directed resections improved survival outcomes (adjusted HR 0.33, 95% CI 0.17–0.64, *p*-0.001), as did primary treatment (adjusted HR 0.43, 95% CI 0.27–0.69, *p* < 0.0001) and presence of lung-only metastases (adjusted HR 0.28, 95% CI 0.08–0.96). Evaluation of those who received multimodal therapy only (*n* = 74) using multivariable analysis found that only metastasis-directed surgery remained significant for survival benefit (adjusted HR 0.47, 95% CI 0.22–0.96, *p* < 0.04), significant for lung resections only (HR 0.18, 95% CI 0.04–0.77, *p* = 0.02).

## 3. Discussion

This retrospective cohort of individuals with metastatic colorectal cancer provides a real-world insight on patient characteristics, overall survival, and treatment patterns in a large cancer centre. Use of multimodal therapies within a multi-disciplinary setting compared to systemic treatment alone demonstrated significant improvement in the survival of individuals with metastatic colorectal cancer in this study. With increasing number of metastases-directed procedures, survival incrementally increased, after adjustment for age, gender, and side of primary tumour. One-year survival in this cohort of patients (79%) is significantly higher than the reported survival data for patients with stage IV colorectal cancer in the United Kingdom (35–45%) [2,15], possibly explained by the increased use of multimodal therapy in this tertiary hospital. Primary tumour resection/primary RT, metastasis-directed surgery, and non-surgical metastasis-directed therapy were all associated with improved survival on univariate analysis. However, evaluation of those patients receiving any multimodal therapy only (and excluding individuals receiving only systemic therapy who were likely considered unsuitable for other therapy) demonstrated a survival benefit of metastatectomies only, confirmed in multivariate analysis. Liver resection, lung resection, and lung ablations conferred a survival advantage, as demonstrated in previous literature [12]; however, undergoing multiple procedures in the same organ did not improve survival outcomes.

Patient selection for multimodal therapy requires a multidisciplinary decision, taking into account fitness, disease burden, the extent of hepatic and extrahepatic disease, requirement for adjunctive chemotherapy, and the potential for R0 resection or complete ablation, with advanced chronological age no longer a contraindication [16]. This is reflected in our series, which showed no heterogeneity of effect by age under or over 70 years. KRAS/NRAS and BRAF mutational status had no effect on survival in this cohort (Table 2), although other literature has demonstrated a negative prognostic effect of BRAF mutations [17]. Not all patients underwent evaluation of mutation status, which may explain this lack of effect. Lung-only metastases have been established as good prognostic factors from previously published literature [18], and this was confirmed in this series of patients. However, the presence of liver-only disease had no prognostic effect.

More than half of patients undergoing resection for mCRC experience a recurrence [19,20]. Studies have shown that this population of patients demonstrate worse outcomes than those without recurrence, although use of repeated interventions is feasible and effective with appropriate patient selection [21,22]. This small series suggests repeated procedures with regional recurrences can be favourable for these patients, and on univariate analysis of site of disease, both liver- and lung-directed resections conferred a survival benefit. However, careful patient and disease-type selection is warranted, particularly as we demonstrated that multiple procedures in the same organ may not have a beneficial effect on survival.

Those patients with three or more interventions had a median survival of 64.8 months, far exceeding the median survival for patients with mCRC overall. This is a reasonable goal for patients with oligometastatic disease suitable for multimodal therapy. Multiple interventions may also reduce the burden of systemic therapy, potentially enabling greater breaks in treatment, and an assessment of the potential effect of this type of management on overall quality of life is required. Many studies have neglected the assessment of patient-reported outcomes following multimodal therapy, although fortunately a number of randomised trials that are currently recruiting include quality of life as a secondary outcome [23,24].

Undergoing resection or radiotherapy of the primary tumour improved outcomes compared to those individuals receiving systemic therapy alone. However, treatment of the primary tumour compared to other multimodal therapies was not significantly associated with improved survival, and therefore the decision to proceed with management of the primary is multifactorial, and should also consider patient symptoms and resectability of metastases [6]. This also raises several other questions that are of ongoing debate, including consideration of synchronous or staged resections of the primary and metastases, or even if primary resection is warranted if metastases are considered unresectable; however, these questions are outside the scope of this study [6].

This series did not show a significant survival benefit of non-surgical metastasis-directed interventions, which included RFA, DEB-TACE, SBRT, and SIRT, when excluding for those patients who underwent systemic therapy only. Possible reasons for lack of survival effect may be explained by our relatively small sample size, and further evaluation of reasons behind patient selection for these treatments are required to better understand the results in this cohort. A randomised trial of radiofrequency ablation plus systemic chemotherapy in patients with non-resectable colorectal cancer has previously demonstrated a progression-free survival benefit of ablation with chemotherapy compared to systemic chemotherapy alone [25]. Long term follow-up confirmed an overall survival benefit in this group, and chemotherapy enabled some patients to convert to resectable disease [26]. Although prospective data remains limited, use of local ablative techniques should be considered as a feasible treatment approach, given toxicity from these procedures are limited; however, our data do not support a survival effect.

We acknowledge the limitations to this study, including the retrospective nature of the study and selected patient population. There is indeed inherent selection bias of patients in whom repeat intervention is feasible, both on the basis of underlying fitness and distribution and disease biology, perhaps with preference for multiple multimodal procedures being performed in those with small-volume, limited-site disease. The main analysis excluded those who received systemic chemotherapy only in order to avoid falsely extrapolating the fact that more treatment modalities are better in these individuals with never-resectable disease, likely extensive disease, and more aggressive disease biology. We also ensured that all consecutive charts of the appropriate study population were selected to try and minimise selection bias, and we adjusted the analysis for age, gender, and side of primary tumour, factors that have previously been shown to have prognostic implications [2,27]. Indeed, there is a risk of immortal time bias in such retrospective analyses, given that OS of patients undergoing one procedure were compared with those undergoing more than one procedure. Univariate *p*-values were not adjusted for multiple testing, as our analyses were exploratory; however, any larger confirmatory study would need to take this into account. A more focused prospective review of patients with oligometastatic disease potentially suitable for multimodal therapy upfront would be beneficial to account for the limitations of retrospective data collection; however, setting up a trial taking into account ethical and logistical dilemmas would be rather challenging. We therefore believe that these data will be crucial in highlighting the importance of a multi-disciplinary approach in management of patients with mCRC. Further studies with a larger sample size comparing cohorts in a district general hospital and tertiary centre experience, such as this, would be useful to determine whether the cost of tertiary centres, where multimodal procedures are more commonly practiced, is justified by improved outcomes.

## 4. Materials and Methods

### 4.1. Patient Population

This retrospective study included all adult patients (≥18 years) with newly metastatic colorectal cancer at University College London Hospital (UCLH) who were treated consecutively from January 2013 to April 2017. Data were collected for patients with histologically confirmed colorectal cancer, with complete treatment details available from the index date (date of first documented metastatic disease) to death or last known follow-up. Patients were ineligible for the study if they did not receive treatment for metastatic CRC within the study period, had malignant tumours other than metastatic CRC, or were those without confirmation of tissue diagnosis.

### 4.2. Data Collection

Patient records were reviewed using the hospital electronic medical records (EMR) for clinico-pathological characteristics, including age, site of the primary tumour, histological subtype, KRAS/NRAS/BRAF mutational status, mismatch repair status, use of neo/adjuvant chemotherapy, number of lines/courses of palliative chemotherapy and chemotherapy type, site of metastatic disease, and type/number of multimodal procedure.

The primary procedure was defined as surgical resection of the primary or for rectal cancers included radical rectal chemoradiotherapy. Surgery for distant metastases was defined as resection of metastases at any disease location. Other non-surgical treatments for localised metastases included thermal ablation (radiofrequency ablation (RFA), microwave ablation (MWA), cryotherapy), drug-eluting bead trans-arterial chemoembolization (DEB-TACE), selective internal radiation therapy (SIRT), and stereotactic body radiation therapy (SBRT). All cases were discussed within a regional specialised colorectal, hepato-biliary, or thoracic MDT at the Royal Free Hospital or the UCLH.

The primary outcome was the impact of multimodal therapy on survival outcomes, compared to systemic therapy alone, and the effect of different types of multimodal therapy on survival outcome.

### 4.3. Statistical Methods

The study endpoint was overall survival (OS). OS was defined as the time in months from diagnosis to either death (as recorded on the hospital EMR) or last follow-up date, and was calculated using the Kaplan–Meier method, with Cox regression analysis used to calculate respective hazard ratios and 95% confidence intervals (CI). The prognostic value of baseline factors was assessed using the multivariate Cox regression model and the log-rank test. *p*-values < 0.05 were considered significant. Univariate and multivariate analyses were performed using STATA (v16.0) (http://www.stata.com). All analyses were adjusted for age, gender, and side of primary tumour (on the basis of known association between right-sided colorectal cancers and worse prognosis). As per the study design, only patients with complete treatment information were included in the study; for other outcomes, the level of missing data was summarised.

## 5. Conclusions

The decision of whether to undertake multimodal treatment of metastatic colorectal primary tumours and secondary metastectomies, in addition to systemic chemotherapy, requires multidisciplinary team input. Careful consideration should be given before carrying out repeat interventions to those with recurrent disease in the same organ. However, in the appropriately selected patient, considering patient fitness, bulk of disease, site and number of metastases, and goals of treatment, multiple sequential procedures may have a significant beneficial impact on patient outcomes.

## Figures and Tables

**Figure 1 cancers-12-03545-f001:**
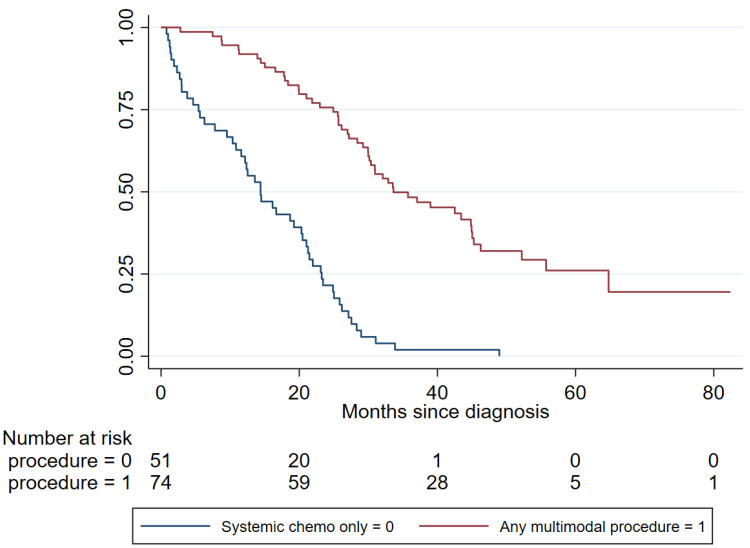
Overall survival of patients undergoing any multimodal procedure compared to those receiving systemic chemotherapy alone.

**Figure 2 cancers-12-03545-f002:**
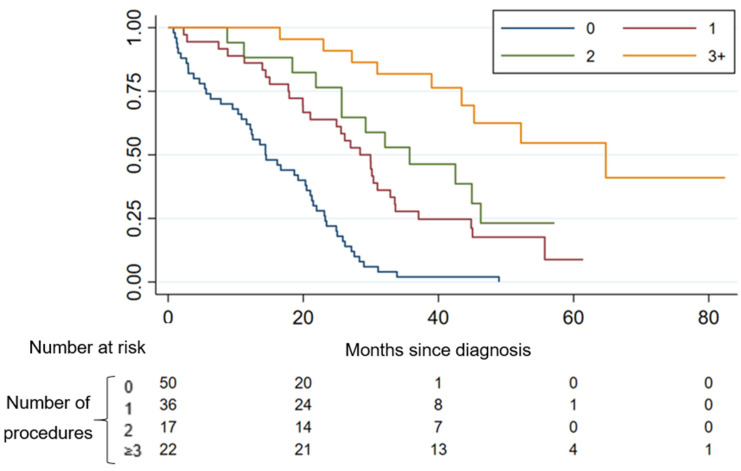
Overall survival by increasing number of multimodal procedures.

**Table 1 cancers-12-03545-t001:** Patient characteristics.

Patient Characteristic	Number (Percentage)
**Gender**	
Male	70 (56%)
Female	55 (44%)
**Age, years**	
Median + IQR	62 (55–68)
Range	19–89
**Side of Primary**	
Left	82 (66%)
Right	43 (34%)
**KRAS Mutation**	
Mutant	48 (47%)
Wild Type	54 (53%)
Missing	23
**BRAF Mutation**	
Mutant	10 (10%)
Wild Type	86 (90%)
Missing	29
**Carcinoembryonic Antigen CEA**	
Elevated (range)	49 (77%) (5.4–28326)
Normal	15 (13%)
Missing	61
**Location of Metastases**	
Liver-Only	55 (44%)
Lung-Only	9 (7%)
Multiple Sites (>1)	61 (49%)
**Adjuvant therapy**	
Yes	23 (180%)
No	102 (82%)
**Neoadjuvant therapy**	
Yes	27 (22%)
No	98 (78%)
**Number of Lines Palliative Chemotherapy**	
0	23 (18%)
1	37 (30%)
2	17 (14%)
3	24 (19%)
≥4	24 (19%)
**Number of Multimodal Procedures**	
0	51 (40.8%)
1	35 (28%)
2	17 (13.6%)
≥3	22 (17.6%)

**Table 2 cancers-12-03545-t002:** Univariate analysis.

Characteristic		Number of Events/Subjects	Adjusted Hazard Ratio (95% CI) **	*p*-Value
Age	≤70 years	102/125	0.94 (0.58–1.52)	0.79
	>70 years	23/125		
Gender	Female	55/125	1.00 (0.67–1.49)	0.99
	Male	70/125		
Sidedness	Right	43/125	0.77 (0.51–1.15)	0.51
	Left	82/125		
Liver-only metastases	No	70/125	0.84 (0.56–1.27)	0.41
	Yes	55/125		
Lung-only metastases	No	116/125	0.195 (0.06–0.60)	0.005
	Yes	9/125		
KRAS/NRAS	Wild-type	46/98	0.79 (0.49–1.27)	0.33
	mutant	52/98		
BRAF status	Wild-type	86/96	1.47 (0.70–3.11)	0.31
	mutant	10/96		
Neo+/−adjuvant chemo received	No	83/125	0.26 (0.16–0.42)	<0.0001
	Yes	42/125		
Systemic chemotherapy only	No	74/125	5.46 (3.49–8.54)	<0.001
	Yes	51/125		
Primary procedure	No	8/74	0.50 (0.23–1.08)	0.08
	Yes	66/74 ^¥^		
Metastatic surgery	No	32/74	0.36 (0.20–0.63)	<0.001
	Yes	42/74 ^¥^		
Liver surgery	No	41/74	0.54 (0.30–0.97)	<0.05
	Yes	33/74 ^¥^		
Lung surgery	No	65/74	0.14 (0.03–0.61)	0.008
	Yes	9/74 ^¥^		
Other non-surgical metastatic procedure *	No	54/70	0.66 (0.33–1.32)	0.24
	Yes	20/74 ^¥^		

* Including radiofrequency ablation (RFA), drug-eluting bead trans-arterial chemoembolization (DEB-TACE), selective internal radiation therapy (SIRT), stereotactic body radiation therapy (SBRT). ** All models include variables age (continuous), gender (male/female), primary tumour side (left/right) by default. All HR are calculated using the top category as the reference level. ^¥^ Analysis excludes those who received systemic chemotherapy only.

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
