# Peer review of "Multimodal Treatment in Metastatic Colorectal Cancer (mCRC) Improves Outcomes—The University College London Hospital (UCLH) Experience"

_cancers, 2020, doi:10.3390/cancers12123545_

Round 1

Reviewer 1 Report

In the present study, Joharatnam-Hogan et al. investigated the impact of multimodal treatment on the survival of mCRC patients. Even though the conceived idea is interesting regarding the low survival of mCRC patients, I have the following major concerns:

  1. The number of patients is relatively small and therefore, it is difficult to evaluate correctly the results. Particularly, the number of patients with only lung metastasis is 9.
  2. A validation cohort is missing from the analysis. I believe that its inclusion could strengthen the results.
  3. Could you explain why the patients who received systematic chemotherapy were excluded from the study?
  4. The section of the discussion is relatively small and does not thoroughly analyze the results.
  5. Please add the number of patients and the events of each stratum in the Kaplan-Meier curves.

Author Response

Response to Reviewers/Editor

We thank the three reviewers for their very useful comments on our manuscript entitled ‘Multimodal treatment in metastatic colorectal cancer (mCRC) improves outcomes- the University College London Hospital (UCLH) experience’. We have addressed each of the reviewer’s comments below.

Reviewer 1:

In the present study, Joharatnam-Hogan et al. investigated the impact of multimodal treatment on the survival of mCRC patients. Even though the conceived idea is interesting regarding the low survival of mCRC patients, I have the following major concerns:

  1. The number of patients is relatively small and therefore, it is difficult to evaluate correctly the results. Particularly, the number of patients with only lung metastasis is 9.

We thank reviewer 1 for their extremely useful comments. We appreciate one of the limitations of the study is its relatively small size (which we have acknowledged in the discussion). We have collected data from all consecutively treated patients over a greater than 4-year period, with the aim of 1) keeping the results reasonably current/relevant to present day management of metastatic colorectal cancer. Therefore, we opted not to review data prior to 2013. 2) Although a limited number of patients with only lung metastases, we believe this reflects real life practice of the range/type/number of patients’ oncologists see with mCRC in their clinics. And therefore, whilst increasing the sample size would undoubtedly be helpful, we believe the cohort of patients presented in this study better reflect current practice.

  1. A validation cohort is missing from the analysis. I believe that its inclusion could strengthen the results.

A validation cohort would certainly be a useful addition. However, we believe this is out of the scope of the current paper and would serve better in a future study. We were limited by the availability of data in our tertiary centre alone but were also keen to present a single centre real-world experience of current practice in London. Future collaboration with other tertiary centres to enable a larger study, with inclusion of a validation cohort, would be a useful future study to consider, and keeping reviewer 1’s suggestion in view we have included this aspect in our discussion (lines 294-297).

  1. Could you explain why the patients who received systematic chemotherapy were excluded from the study?

The initial analysis compared those who received systemic chemotherapy alone with those who received multimodal therapy and systemic chemotherapy. However further analysis excluded those who received systemic chemotherapy alone deliberately, as the main purpose of the study was to evaluate outcomes in those receiving multimodal therapy. Patients who receive systemic chemotherapy alone represent a distinct biological cohort, likely to have more extensive/aggressive, irresectable disease, and therefore inclusion of these patients would lead to selection bias and false extrapolating of the results that more treatment modalities are better. We have now addressed this more explicitly in the discussion (lines 281-282).

  1. The section of the discussion is relatively small and does not thoroughly analyze the results.

We have reviewed the discussion section and have now added additional paragraphs/sentences to analyse the totality of the results (lines 234/5, 243/4, 255-262)

  1. Please add the number of patients and the events of each stratum in the Kaplan-Meier curves.

The number of patients/events have already been included in the tables under each Kaplan-Meier curve.

Reviewer 2 Report

This study presents experience of multi-modal therapy for mCRC patients at a single hospital. Overall the manuscript is well written. The major limitation is the strength of evidence as it is a single-centre retrospective analysis based on observational data.   Major comments: 1. There have been previous larger studies that reported the benefit of multimodal therapy for mCRC. How does this study compare to existing evidence? 2. Have you considered the problem of multiple testing? A large number of variables were tested in this study. 3. How many patients died and how many of them were alive until the last follow-up? These are important numbers which ought to be reported. 4. I wonder why the authors reported p values from log-rank test along with HRs and CIs estimated from multivariable Cox model? The Cox model tests the significance of hazard ratios, and these p values should be reported with the effect sizes. The log-rank p values may be reported alongside the KM curves.   Minor comments: Abstract: I rarely see bibliographies in the abstract of a paper. Consider removing all references in this part. Please ignore this comment if the journal requests otherwise. Table 2: column 3 is not clear. What is the number of events? Is it >=70 years for age? Please clarify it a bit more. The word "appropriate" is vague in the conclusion. Consider clarifying it.

Author Response

This study presents experience of multi-modal therapy for mCRC patients at a single hospital. Overall the manuscript is well written. The major limitation is the strength of evidence as it is a single-centre retrospective analysis based on observational data.  

We appreciate reviewer’2 positive comments, but also the major limitation, which we have noted in our discussion.

Major comments:

  1. There have been previous larger studies that reported the benefit of multimodal therapy for mCRC. How does this study compare to existing evidence?

Most studies in the literature to date evaluate single procedures or single organs, such as liver or lung directed resection. Our study evaluates the totality of all procedures used in the real world setting of metastatic colorectal cancer.

  1. Have you considered the problem of multiple testing? A large number of variables were tested in this study.

We agree that multiple testing can reduce the strength of the study. However, the variables analysed are all known to impact outcomes in metastatic colorectal cancer, and so spurious p-values are less of a concern. We also believe that the use of multivariate analysis strengthens our findings. However, we have added this issue to the limitations section of the discussion (lines 287-289) based on Reviewer 2’s recommendation.

  1. How many patients died and how many of them were alive until the last follow-up? These are important numbers which ought to be reported.

At the time of follow up 81% (101/125) had died, and the remaining 24/125 were alive and censored. We have now included this detail in the results (145-146).

  1. I wonder why the authors reported p values from log-rank test along with HRs and CIs estimated from multivariable Cox model? The Cox model tests the significance of hazard ratios, and these p values should be reported with the effect sizes. The log-rank p values may be reported alongside the KM curves.  

We have updated the wording in the manuscript to clarify that log-rank p value was used when comparing the 4 survival curves in figure 2 and have removed log-rank when reporting the whole HR with CIs elsewhere.

Minor comments:

  • Abstract: I rarely see bibliographies in the abstract of a paper. Consider removing all references in this part. Please ignore this comment if the journal requests otherwise.

We have now removed these references in response to reviewer 2’s comment.

  • Table 2: column 3 is not clear. What is the number of events? Is it >=70 years for age? Please clarify it a bit more.

We have now added clarity to column 3 of table 2 and have added a footnote to the table that ‘All HR are calculated using the top category as the reference level,’ to clarify what the number of events is referring to (line 182).

  • The word "appropriate" is vague in the conclusion. Consider clarifying it.

We agree that the wording is currently vague, and therefore we have re-worded the conclusion to enhance clarity (lines 335-342).

Reviewer 3 Report

The Authors analized the clinico-pathological characteristics, multimodal treatments and outcomes of all patients with mCRC treated consecutively between January 2013 and April 2017 and a minimum follow up period of 30 months at University College London Hospital.

They accurately identify and discuss in the "limitation" section several bias of their study including retrospective nature of data, but underline that a prospective trial however, setting up a trial could taking into account ethical and logistical dilemmas would be rather challenging, then they concluded that further studies with a larger sample size comparing cohorts in a district general hospital and tertiary centre experience would be useful to determine whether the cost of tertiary centres, where multimodal procedures are more commonly practised, is justified by improved outcomes.

They concluded that a careful consideration should be given before carrying out repeat interventions to those with recurrent disease in the same organ. However in the appropriate patient, multi-disciplinary team input on the use of multiple sequential procedures may have a significant impact on patient outcomes.

The study is well conducted, the authors highlight the limitations inherent in the retrospective nature of the case series, and with their statistics and serious criteria for inclusion and exclusion they tried to limit the bias.

Table 1:

ADJUVANT THERAPY        YES  23 (180%) (correct: 18%)
                                      NO  102 (82%)

Author Response

Reviewer 3:

The Authors analized the clinico-pathological characteristics, multimodal treatments and outcomes of all patients with mCRC treated consecutively between January 2013 and April 2017 and a minimum follow up period of 30 months at University College London Hospital. They accurately identify and discuss in the "limitation" section several bias of their study including retrospective nature of data, but underline that a prospective trial however, setting up a trial could taking into account ethical and logistical dilemmas would be rather challenging, then they concluded that further studies with a larger sample size comparing cohorts in a district general hospital and tertiary centre experience would be useful to determine whether the cost of tertiary centres, where multimodal procedures are more commonly practised, is justified by improved outcomes. They concluded that a careful consideration should be given before carrying out repeat interventions to those with recurrent disease in the same organ. However in the appropriate patient, multi-disciplinary team input on the use of multiple sequential procedures may have a significant impact on patient outcomes. The study is well conducted, the authors highlight the limitations inherent in the retrospective nature of the case series, and with their statistics and serious criteria for inclusion and exclusion they tried to limit the bias.

We appreciate reviewer 3’s positive comments about the strength of the study, how well conducted it is and the ways we have tried to minimise bias.

Round 2

Reviewer 1 Report

Most issues raised by the Reviewers after the first submission of this manuscript were properly addressed. The low number of patients, and especially those with lung metastasis, is still a major limitation of the current study; however, the authors have provided a reasonable explanation and discusses this issue in the revised version. Overall, the manuscript has been improved.

Reviewer 2 Report

The authors have addressed my comments.